# Working towards an ERAS Protocol for Pancreatic Transplantation: A Narrative Review

**DOI:** 10.3390/jcm10071418

**Published:** 2021-04-01

**Authors:** Madhivanan Elango, Vassilios Papalois

**Affiliations:** Department of Surgery and Cancer, Imperial College London, London SW7 2AZ, UK; vassilios.papalois@nhs.net

**Keywords:** kidney, pancreas, transplantation, enhanced, recovery

## Abstract

Enhanced recovery after surgery (ERAS) initially started in the early 2000s as a series of protocols to improve the perioperative care of surgical patients. They aimed to increase patient satisfaction while reducing postoperative complications and postoperative length of stay. Despite these protocols being widely adopted in many fields of surgery, they are yet to be adopted in pancreatic transplantation: a high-risk surgery with often prolonged length of postoperative stay and high rate of complications. We have analysed the literature in pancreatic and transplantation surgery to identify the necessary preoperative, intra-operative and postoperative components of an ERAS pathway in pancreas transplantation.

## 1. Introduction

Enhanced recovery after surgery (ERAS) protocols have revolutionised the field of surgery with significant improvements in postoperative length of stay (LOS), healthcare cost and patient satisfaction [1,2,3]. It is an evidence-based, multidisciplinary multimodal approach with surgeons, anaesthetists, nurses, and other allied health professionals all playing a role to improve the quality of perioperative care of the patient. Protocols have been established in renal transplantation surgery [4,5,6,7] and pancreatic surgery [8,9,10,11,12,13], but as of December 2020, a literature search yielded no published work on ERAS protocols post simultaneous pancreas-kidney (SPK) transplantation, pancreas after kidney (PAK) transplantation or pancreas transplantation alone (PTA). A large portion of ERAS protocols are very similar across many surgeries: preadmission medical optimisation, minimal drains, early ambulation and post-discharge follow up. In this narrative review, we aim to highlight some of the unique aspects that must be considered when starting an ERAS protocol for pancreas transplantation.

## 2. Why Is an ERAS Pathway So Important for Pancreatic Transplantation?

Pancreas transplantation is known to confer high patient morbidity and mortality, often leading to complications with a prolonged hospital stay. The median length of stay after transplant varies by centre but has been reported in the range of 8–27 days [14,15], with some studies putting the postoperative complication rate at close to 40% with reoperation rates exceeding 30% [16,17]. The patients themselves are also known to be high risk, with virtually all patients have diabetes mellitus, and many of these patients have concomitant cardiovascular or renal disease. Studies have shown that diabetes and renal disease are independent risk factors for predicting poor postoperative outcomes [18,19,20]. This high-risk surgery on high-risk patients needs an ERAS pathway, now more than ever.

## 3. Preoperative ERAS Components

### 3.1. Preoperative Education

Preoperative counselling regarding diet, smoking, alcohol, and weight loss is known to improve surgical outcomes in other surgical specialities while allowing the patient to take ownership of their health. Some improved outcomes have been reported in kidney transplantation, SPK transplantation and pancreatic surgery; we argue that the biological plausibility and strong evidence in other surgical fields lend support to the inclusion of preoperative counselling in an ERAS protocol for pancreatic transplantation.

We firstly argue that all pancreas transplant candidates who smoke should undergo smoking cessation counselling at their pre-transplant evaluation to reduce both short-term and long-term complications post-transplant. While there are currently no studies on the effects of smoking in pancreatic transplantation, we have formed this hypothesis mainly based on studies in smoking in kidney transplantation, liver transplantation, pancreaticoduodenectomies and large meta-analyses looking at multiple fields of surgery. Large meta-analyses of both observational and randomised controlled trials (RCTs) have shown that preoperative smoking cessation with behavioural and pharmacological interventions have led to a decrease in postoperative complications, namely pulmonary complications and wound infection rates [21,22,23,24]. They find that the ideal length of smoking cessation is more than four weeks preoperatively, and Mills et al. found an increase in effect size of 19% per week of smoking cessation [21]. These interventions are likely effective, with a recent RCT involving kidney transplant recipients showing a decrease in exhaled carbon monoxide (an objective marker of smoking) due to smoking cessation advice and carbon monoxide measurement. In other fields of surgery, studies have shown a long term benefit of preoperative counselling that is long-lasting and effective [25,26,27].

Within kidney transplant recipients, multiples studies have demonstrated an increased risk of mortality, graft failure and malignancy in smokers compared to never smokers. Agarwal et al. found increased mortality and graft failure rates in renal transplant recipients who continued to smoke post-transplant compared to never smokers. Past smokers, on the other hand, showed increased mortality compared to the never smoker group but no significant increase in graft failure, indirectly supporting the idea that smoking cessation reduced graft failure [28]. Several other studies corroborate this finding that graft failure risk reduced to baseline post smoking cessation [29,30,31]. Indeed, there is an increased short term risk of rejection by day 10 post kidney transplant (adjusted HR, 1.8; 95% CI, 1.10–2.94; *p* = 0.02) in ever smokers compared to never smokers [32], as well as an increased risk of early readmission (<30 days post-discharge) in smokers [33]. Smoking is also associated with post-transplant malignancy, and cardiovascular events post kidney transplantation [34,35]. Within liver transplantation, active smokers at the time of transplant had an increased risk of mortality from cardiovascular causes and sepsis [36], with smoking cessation before transplant associated with increased survival [37]. Most interestingly, there was an increased length of hospital stay (13.4 vs. 7.9 days; *p* = 0.02) and increased cost of admission ($129,185 vs. $99,694; *p* = 0.02) in smokers vs. non-smokers [38]. In pancreaticoduodenectomies, smoking is an independent risk factor for pancreatic fistula development as well as an independent predictor for discharge to a non-home environment (rehabilitation, skilled care, or acute care facilities) [39,40]. Some centres report 30% of SPK recipients develop a pancreatic fistula, with 31% of these requiring a relaparotomy; length of stay was also longer by eight days in the fistula group, but this was not statistically significant [41]. We recognise that definitive proof would require prospective studies on preoperative counselling to look at both the short term and long-term outcomes of SPK recipients. Nevertheless, we believe the wealth of evidence demonstrating the increased risks of smoking in similar patient populations both in the initial admission and the long term justify the place of preoperative smoking cessation in an SPK/PTA ERAS protocol.

### 3.2. Prehabilitation and Weight Loss

Obesity has been shown in multiples studies to be associated with negative outcomes in pancreatic transplantation. There is an increased risk of graft loss, mortality, wound dehiscence, hernia formation, necrotising fasciitis and relook laparotomy [42,43,44,45], with Hanish et al. finding a much-increased rate of postoperative complications in obese SPK recipients (OR 6.8; *p* < 0.001) [42]. Furthermore, some studies have found decreased graft and patient survival in obese patients [46,47]. Despite this finding, Owen et al. found poor sensitivity and specificity when trying to use BMI to predict these outcomes. The area under the curve (AUC) for the receive-operator characteristic (ROC) curves ranged from 0.50–0.55, suggesting that a BMI cut-off cannot be used to predict which patient will have poor outcomes post-transplantation. However, a patient’s weight can still be optimised to potentially improve their individual post-operative outcomes.

With a clear association between obesity and postoperative complications, studies are emerging on the use of prehabilitation before surgery to increase aerobic capacity and possibly reduce weight. An RCT on using a prehabilitation program (with endurance training and promotion of physical activity) in patients undergoing elective major abdominal surgery showed a halving of postoperative complications (RR, 0.5; 95% CI, 0.3–0.8; *p* = 0.001) [48]. Length of stay in prehabilitation vs. control was 13 days vs. eight days in the hospital and four vs. one in the intensive care unit (ICU), respectively. These results were not statistically significant, but the study was not adequately powered to detect these differences. While we recognise that some patients will have a history of amputations, reduced visual acuity and neuropathies that will preclude meaningful participation in an exercise program, many candidates will still meet the criteria for prehabilitation as they spend significant time on the waiting list [48]. A pilot prehabilitation study in dialysis patients demonstrated an improvement in physical activity [49]. In those patients that went on to have a kidney transplant, their length of stay was 31% shorter compared to controls matched for age, sex, and race (RR = 0.69; 95% CI, 0.50–0.94). Crucially only five patients went on to receive a transplant in this observational study highlighting the need for prospective studies to confirm this finding. Given the high prevalence of obesity in pancreas transplant recipients, the demonstrated association between obesity and short-term complications and the evidence of prehabilitation in similar abdominal surgeries, we suggest that there is a benefit to including prehabilitation in the ERAS pathway for pancreas transplantation while prospective studies fully examine its benefit.

### 3.3. Preoperative Cardiac Assessment

Cardiovascular events are the major cause of mortality in patient with end-stage renal failure and are a major cause of postoperative morbidity in pancreas transplant patients [50,51,52,53]. As organ recipients become older and more overweight, their cardiovascular risk profile increases. There is known to be a high prevalence of coronary artery disease (CAD) in diabetic patients, and this can lead to perioperative complications. One study quotes the risk of cardiovascular events at around 10% post-SPK transplantation, with almost half of these in the perioperative period [52]; it is also known that the mortality from perioperative myocardial infarction is around 3–25% [54]. It is therefore important to assess high-risk patient preoperatively so that they can receive the appropriate intervention.

Initially, most candidates will undergo transthoracic echocardiography (TTE) to determine ejection fraction [55]; those with poor ejection fraction are likely to be poor transplant candidates. Fortunately, most transplant candidates have an adequate systolic function. We must then determine which patients are at high risk to select candidates who are most likely to benefit from further testing, including invasive coronary angiography, which is associated with its own risks. While there are no specific criteria for pancreas transplant candidates, the American College of Cardiology (ACC) has released guidelines for potential kidney transplant recipients [56]. They suggest patients with at least three of the following risk factors should undergo further testing: “diabetes mellitus, prior cardiovascular disease, >1 year on dialysis, LV hypertrophy, age > 60 years, smoking, hypertension, and dyslipidemia” [56]. While this has not been validated in a pancreas transplant recipient population, it remains a good starting point for evaluating cardiovascular risk. In those at high risk requiring further testing, there is a choice of functional non-invasive imaging, e.g., dobutamine stress echocardiography (DSE), anatomical non-invasive imaging, e.g., CT coronary angiography (CTCA) and invasive coronary angiography to look at perfusion of the heart. DSE is a screening test and can predict CAD in renal transplant candidates with sensitivity and specificity of 88% and 94%, respectively [57]. Given the high pre-test probability of these patients having CAD (47% in one study [58]), the post-test probability of CAD despite a negative DSE using the above values is 10%. Therefore, these patients will need to undergo further testing by CTCA to rule out significant CAD. Some techniques looking at the presence of calcification in the aorta using lateral spine X-rays are known to be associated with post-SPK transplantation mortality and graft outcome [59]. However, with such a high prevalence of the disease, more definitive testing is therefore needed to rule out CAD, which can be in the form of CT calcium scoring or a CT coronary angiogram (CTCA) [58]. Patients on haemodialysis are known to have a high calcium score, so CTCA is preferred for these patients. Therefore, we propose that patients who are not low risk (potentially by the above ACC criteria) undergo functional and non-invasive anatomical imaging with more work needed to validate criteria for pancreas transplant candidates with low cardiovascular risk. However, we need to highlight that there is an ongoing debate regarding the choice between CTCA and invasive coronary angiography when it comes to anatomical imaging that depends on local expertise and individual risk of CAD.

Following this, patients can undergo revascularisation with stenting or bypass grafting as appropriate. CABG is often preferred in this patient group, and multiple RCTs have demonstrated the superiority of CABG in diabetic patients [60,61,62]. Patients who receive coronary stents are also likely to take clopidogrel which can complicate any further surgery. In a single centre, 7% of patients underwent CABG before pancreas transplantation, and 5% of patients underwent CABG based on the results of their pre-transplant evaluation [58]. Preoperative cardiac screening in this patient subset is picking up a high number of patients with clinically significant CAD; patient must be screened for CAD as part of any preoperative assessment, and some patients may need repeat screening if they are on the waiting list for a particularly long time.

### 3.4. Preoperative Vascular Assessment

The vessels of the pancreatic graft are anastomosed to the recipient’s iliac vessels. It is known that calcification of these vessels can increase graft loss and even mortality risk post-transplantation [63,64]. Unexpectedly finding a calcified vessel intra-operatively may delay the ischemic time of the graft organ and the length of the operation. Therefore it is beneficial to have some form of imaging of the iliac vessels preoperatively to guide the operative procedure and allow close monitoring of those at high risk postoperatively. There are two main methods of assessing vascular calcification in the iliac arteries: pelvic X-ray and computerised tomography (CT) without contrast. One study used pelvic X-ray as a screening tool before progressing to CT in high-risk patients preoperatively; in 24% of patients, the imaging helped determine the side of surgery, showing its potential importance in an ERAS pathway [65]. However, there is little data on the sensitivity of pelvic X-rays for determining significant calcification, and given the relative ease of obtaining non-contrast CT imaging, an unenhanced is certainly the best option for evaluating the iliac arteries.

## 4. Intra-Operative ERAS Components

### 4.1. Antibiotics Prophylaxis

Surgical site infection (SSI) is a major cause of morbidity post-transplantation, with some series reporting an incidence of up to 45% [66]. Due to the immunosuppression required for pancreatic transplantation along with their co-morbidities, transplantation patients are particularly prone to acquiring SSI. Studies in major surgeries have shown that infections can double the LOS, double the mortality, increase ICU time by 60%, double the cost of stay and significantly increase the chance of readmission [67,68]. With any major surgery, appropriate antibiotic prophylaxis is known to reduce the risk of SSI, and this is especially true within the field of transplantation [69,70,71].

In pancreas transplantation, Staphylococci and Enterococci are known to be the primary source of SSI with less common causes, including gram-negative organisms [66]. As such, the choice of antibiotic should be tailored to these pathogens, and first-generation cephalosporins are an ideal choice. There is some evidence that vancomycin reduces initial LOS in pancreas transplant recipients, but as this was a small RCT with only 24 patients conducted in the 1990s, more research is needed before this can be recommended [72]. The exact choice of antibiotic should be determined, taking into account the local resistance patterns and antibiograms [73,74]. In a small observational study, a reduction in the incidence of *Candida* SSIs was associated with fluconazole prophylaxis [66]. This potentially justifies the use of a single dose of fluconazole perioperatively with longer courses for patients with specific risk factors for fungal infection. Finally, we recommend a short course of antibiotics lasting at most 96 h. The STOP-IT (Short-Course Antimicrobial Therapy for Intraabdominal Infection) trial compared a fixed four-day course of postoperative antibiotics to antibiotics until two days post-resolution of fever, leucocytosis and ileus [75]. The median length of antibiotic use was four days vs. eight days respectively. There was no significant difference in either SSI incidence, intra-abdominal infection or death suggesting that in the absence of a specific reason, postoperative antibiotics should be limited to four days.

### 4.2. Operative Techniques

Operative time is an important factor in all fields of surgery, and it is known to relate to postoperative outcomes and incidence of SSI [76,77]. Within transplantation surgery, both cold ischemic time and warm ischemic time are known to impact graft outcomes, with prolonged cold ischemic time associated with reduced pancreatic graft survival [78,79,80]. It is therefore imperative to take all steps to reduce cold ischemic time and operative time to reduce short term postoperative complications and improve long term graft outcomes.

This means that simple measures, including the selection of local donors, preparation of the back table with all necessary instruments and available staff to work on the donor graft as soon as it arrives, could have an impact on outcomes. Each centre will have its own preferences for the back table vascular preparation, duodenal preparation and splenic dissection. There is some evidence that reconstruction of the gastroduodenal artery (GDA) improves perfusion of the pancreas [81,82,83]; Socci et al. found that triple arterial revascularisation with an interposition graft using the donor iliac vessels was associated with lower rates of postoperative gastrointestinal (GI) bleeding (a marker of insufficient perfusion of duodenojejunal anastomoses) and GI bleeding was associated with worse patient and graft survival [84]. While this is a retrospective study, it highlights the possible effects of back table choices, which should be investigated further with prospective studies.

There have also been studies looking at the position of the pancreas and kidney in SPK and PAK transplantation. Traditionally, the pancreas has been placed on the contralateral side of the kidney due to concerns that the proximal pancreatic graft may potentially jeopardise an ipsilaterally placed kidney. However, two retrospective studies in SPK transplantation [85,86] and one in PAK transplantation [87] looking at ipsilateral organ placement do not demonstrate the superiority of one technique over the other. Interestingly, Papachristos et al. found that ipsilateral SPK transplantation (iSPK) resulted in significantly less operative time than contralateral SPK (cSPK): 293 min vs. 359 min (*p* = 0.04), although there was no difference in cold ischemic time. They also found that iSPK recipients were more likely to be female (72%), while cSPK recipients were more likely to be male (90%). The authors hypothesise that the wide female pelvis allows easier implantation of the kidney on the ipsilateral side. Given the lack of evidence suggesting the cSPK transplantation is superior, we argue that the surgical team should choose the laterality of transplant pre-operatively based on donor factors (organ size) and recipient factors (iliac vessel quality, pelvic size, and BMI). More work should be undertaken to look at the outcomes of iSPK transplantation with propensity matching before we can definitively include this in an ERAS pathway.

Importantly, the back-table procedure should be standardised within a centre to allow for efficient preparation and minimisation of cold ischemic time. The use of automated staplers for duodenal work may also help to reduce operative time and should be pursued as part of the ERAS pathway. Porcine studies in renal transplantation have demonstrated the use of an automatic vascular anastomotic device, initially designed for coronary artery bypass grafting, can reduce warm ischemic time and may play a beneficial role in pancreas transplantation [88]. Finally, robotic pancreas transplantation is being developed for high-risk obese patients at higher risk of postoperative complications [89,90,91,92]. While these novel techniques should not be routinely used in pancreas transplantation without more evidence and training, they represent an exciting opportunity for further future improvements in surgical outcomes.

### 4.3. Enteric vs. Bladder Drainage

Pancreas transplantation is mainly performed for the endocrine function of the pancreas. While pancreas transplant recipients often have adequate exocrine function, the exocrine secretions of the graft must still be drained. In the early days of transplants, bladder drainage was used as it allowed for easy monitoring of urinary amylase as a marker of rejection [93,94]. Early rejection can often be treated, but once hyperglycaemia is present, there is a low probability of reversal. Bladder drainage also avoided the risks of abdominal contamination caused by an enterotomy and Roux-en-Y loop formation. However, bladder drainage is associated with urinary tract infection, metabolic abnormalities (hyperchloremic metabolic acidosis secondary to bicarbonate loss through the urine), and recurrent episodes of graft pancreatitis due to urinary reflux [93]. Dehydration, in particular, sometimes requires hospital visits, and bladder drainage was associated with more day unit visits for treatment of acidosis and dehydration in one study [95]. As surgical techniques have improved, enteral drainage is becoming more prevalent, making up more than 90% of pancreas transplants from 2006–2016 [94]. Multiple retrospective analyses have shown very little difference in outcomes between the two types of drainage with similar survival rates. There is some evidence that graft survival rates and acute rejection rates are higher in enteral drainage, but these are seen in retrospective studies with a high degree of bias; in one study, there was a significantly higher degree of HLA mismatch in the enteral drainage group which could explain the increase in rejection [96]. It is also known that up to 22% of transplantations with bladder exocrine drainage end up being converted to enteric drainage due to complications; these re-operations are likely to prolong initial hospital stay or require readmission to hospital [97]. As there are similar survival rates for both techniques with bladder drainage resulting in unique postoperative complications and high rate of conversion to enteric drainage, we suggest that enteric drainage of some form should be part of the ERAS pathway.

### 4.4. Analgesia

Effective analgesia is the cornerstone of any postoperative course and is crucial to ensuring a patient has a timely discharge while minimising the risk of postoperative complications. There has been a general shift to reducing opioid use due to their negative effects on the GI system and the opioid pandemic in the United States [98,99]. Opioids are known to cause postoperative nausea and vomiting, both of which are associated with delaying oral consumption [100]. SPK patients may also have end-stage renal disease (ESRD), limiting their ability to take certain opioids due to the accumulation of metabolites [101]. Pancreas transplant recipients often require increased anticoagulation due to the risk of allograft thrombosis, sometimes limiting the epidural options available [102,103,104]. Epidural analgesia is also known to increase the LOS while only providing superior pain control on the day of surgery in pancreatic surgery [105].

New analgesic options include transversus abdominis plane (TAP), rectus sheath and quadratus lumborum blocks with an infusion of local anaesthetic [106,107,108,109,110]. Hausken et al. showed that a rectus sheath block could provide equivalent analgesia to thoracic epidural analgesia in pancreas transplantation [104]. In a double-blind RCT comparing IV morphine PCA (patient controlled analgesia) vs. IV morphine PCA with TAP block vs. IV morphine with TAP block and dexmedetomidine in renal transplant patients, Yang [111] et al. found that both TAP block groups had reduced pain, decreased morphine consumption and decreased nausea and vomiting compared to the IV morphine PCA group. They found that the addition of dexmedetomidine reduced morphine requirements and improved pain control compared to a TAP block with only ropivacaine. Another retrospective study comparing IV opioids vs. TAP catheter vs. TAP block with liposomal bupivacaine in pancreas transplantation found that TAP blocks were associated with less IV opioid use and faster return of bowel function with a block and liposomal bupivacaine proving more effective than a TAP catheter [112].

While these studies did not find an effect in the LOS, they demonstrated that adequate analgesia could be provided with local blocks with a reduction in opioid use and faster return of bowel function (ROBF). These studies indicate the need to include a block of some form in pancreas transplant patients in preference to IV opioid PCA. Kolacz et al. found that a quadratus lumborum block might be more effective than a TAP block in renal transplant recipients showing that the exact anatomical location and anaesthetic agent with or without adjunct of the block are still up for debate with local factors including anaesthetic skills needing consideration [113]. However, pancreas transplant centres should move towards the utilisation of local blocks in an ERAS protocol with further work in prospective studies.

It is also important not to forget simple adjuncts in the use of pain control. These may allow the patient to experience less pain as well as facilitating reduced opioid usage. Regular intravenous acetaminophen has been shown in multiple RCTs to reduce post-operative pain, reduce opioid usage as well as decrease delirium [114,115,116]. It has minimal side effects, so we can recommend its routine use post-pancreas transplantation. More work is needed to study its efficacy in this population, as well as the relative efficacy of oral vs. intravenous acetaminophen. There are also multiple randomised controlled trials showing the benefits of a 5% topical lidocaine patch in surgical patients [117,118,119]. Interestingly, Fiorelli et al. found that pre-emptively using a lidocaine patch before thoracotomy resulted in decreased opioid use. We recommend the use of local blocks with an infusion of local anaesthetic, so it is not clear if a lidocaine patch would provide an additional benefit. More research should look into the combination of local blocks with the addition of a lidocaine patch before this practice is widely adopted.

### 4.5. Nasogastric Intubation

In the past, the use of nasogastric (NG) tube prophylactically in patients undergoing major abdominal surgery was intended to reduce anastomotic complications, prevent gastric distention, increase patient comfort and speed up the return of bowel function [120]. However, a large meta-analysis and a Cochrane review have found the opposite: NG intubation is associated with a slower return of bowel function and increased LOS [120,121]. In a retrospective study on pancreas transplant patients, avoiding routine NG tube placement was found to have no impact on survival or graft outcomes but resulted in a significantly reduced LOS. Therefore, we recommend against the routine placement of NG tubes and suggest that they only have a role in the case of established postoperative ileus.

### 4.6. Fluid Therapy

Intravascular fluid status plays an important role in patient care; the hypovolaemic patient is at risk of hypoperfusion and organ dysfunction, while the hypervolemic patient is at risk of tissue oedema and adverse outcomes [122]. Retrospective studies have shown that in pancreaticoduodenectomy (PD) patients, a higher amount of fluid therapy is associated with increased perioperative complications and increased length of stay [123,124]. However, these retrospective studies are at a high risk of bias as a patient with higher rates of complication may require more fluid therapy. Multiple RCTs have recently been carried out looking at the effects of a restrictive fluid regimen for PD patients. These restrictive fluid regimens are often based on goal-directed fluid therapy (GDFT); fluid is administered in response to a particular cardiovascular parameter such as central venous pressure, cardiac index or stroke volume. A recent meta-analysis of these trial by Chen et al. found no significant difference in post-operative outcomes [125]. However, they note that there was large heterogeneity in the definitions for restrictive and standard fluid therapies, so they advised caution when drawing conclusions from their paper. Another meta-analysis looking at restrictive fluid therapies in elective major abdominal surgery patients did find a decrease in morbidity, LOS, ICU LOS, and time to passage of faeces. Interestingly, they found that in patients enrolled on an ERAS pathway; there were only significant decreases in ICU LOS and time to passage of faeces. It is difficult to recommend GDFT when multiple RCTs have shown minimal if any benefit to this time and labour intensive process. However, there have been no RCTs or prospective studies investigating GDFT in pancreas transplant patients. Given the association of excess fluid with adverse outcomes and lack of any demonstrable harm with fluid therapy, we recommend that excess fluid administration be avoided with more work needed to see if GDFT may be beneficial in pancreatic transplantation.

## 5. Anti-Coagulation

Thrombosis is a feared complication of pancreatic transplantation and the most common cause of early allograft loss, with its prevalence ranging from 3% to 20% [102,103,126,127,128]. A large systematic review reported that graft loss is seen in 83% of cases with thrombosis [129], often necessitating prolonged hospital stay. Once the condition has been diagnosed with CT imaging or ultrasound, it is often too late, so many therapies focus on prevention with anticoagulation. There is no standard consensus regarding the optimal prophylactic anticoagulation, and most centres have their own protocols. A Japanese centre has reported a 3.8% thrombosis rate without the use of heparin, while an analysis of the United Network for Organ Sharing (UNOS) database and the International Pancreas Transplant Registry (IPTR) between 2014 put the equivalent rate between 4.1% and 6.4% including patients who had heparin [127,128]. There are a wide variety of definitions of thrombosis used across the literature, which differ from arterial vs. venous to clinical vs. subclinical (only identified on imaging with limited clinical significance) [130]; this makes a direct comparison between different protocols challenging and should be the topic of a systematic review by itself. In this review, we will briefly discuss some components that should be included in an ERAS pathway, but the details will need to be centre specific, based on their patient demographics and will need to be validated in prospective trials.

It is known that the low flow, low-pressure nature of pancreata predisposes to thrombosis. These patients, by virtue of their disease processes and surgery, are also in a hypercoagulable state, with 18% of patient with allograft thrombosis having inherited coagulopathies (Factor V Leiden and Prothrombin gene mutations) [131,132]. In patients with a significant degree of renal disease, there may also be a degree of uraemic thrombocytopathy present complicating matters [131,133,134,135]. Donor factors (including age, weight, and gender) and perioperative factors (cold ischemic time, warm ischemic time, preservation solution, exocrine drainage, and arterial reconstruction) are known to affect thrombosis risk [103,129,135]. There is also evidence in retrospective studies that anticoagulation does prevent allograft thrombosis in these patients, but this must be balanced with the risk of rebleeding and risk of repeat exploratory laparotomy caused by excess anticoagulation [129,131,132,135,136,137,138,139,140]. For each case, an assessment of thrombotic risk should be made based on donor factors, recipient factors and perioperative factors, and a degree of anticoagulation should be begun commensurate with this calculated risk. As mentioned, we will highlight areas that we feel should be included in this process, but the full anticoagulation protocol will need to be verified in prospective studies:All patients should be offered preoperative hypercoagulability screening due to the high prevalence of hypercoagulable mutations in patients with allograft thrombosis compared to the general population [131]. These patients will need a particularly effective anticoagulation regimen.Regular preoperative, intra-operative and postoperative monitoring of coagulation through thromboelastography (TEG). TEG allows rapid, real-time assessment of both clot formation as well as fibrinolysis making it ideal for titration of anticoagulation in the perioperative period [141,142]. In a retrospective study, Gopal et al. demonstrated that there was reduced bleeding (18% vs. 45%, *p* = 0.05) and reduced length of hospital stay (18 vs. 31 days, *p* = 0.03) in anticoagulation titrated using TEG vs. standard coagulation tests (prothrombin time and activated partial thromboplastin time), respectively [137].Regular preoperative monitoring of platelet function through the collagen-epinephrine closure time (Col/Epi) and collagen-ADP assay (Col/ADP) could be beneficial, but more work is needed to evaluate their clinical value. Raveh et al., in their first of a kind study, assessed the impact of these platelet function tests on pancreatic allograft thrombosis; they found a strong association between platelet dysfunction and both allograft thrombosis and venous thrombosis severity score [135]. Lower arterial thrombosis severity scores were found in patients on aspirin, and abnormal platelet function assays were also found to be independently predictive of graft thrombosis in a multivariate analysis.Calculation of thrombosis risk from the donor, perioperative and recipient factors- systematic reviews and observations studies have identified multiple factors that may be associated with allograft thrombosis as mentioned before [103,129]. Each one by itself may not have a large effect, but their combination may greatly increase the risk of thrombosis. Raveh et al. proposed a minor scoring system for combining these factors, and this should be further improved with local factors [135].Imaging and grading of potential allograft thrombosis should be standardised per centre to ensure that anticoagulation is only increased when it likely to clinically benefit the patient. The large heterogeneity in thrombosis rates can be partially explained by the varying definition, with subclinical thrombi being included in these rates. For lower grades of arterial and venous thrombi, Hakeem et al. demonstrated that there were similar patient and graft survival in those therapeutically anticoagulated vs. conservative management with standard anticoagulation [130]. Reexploration for bleeding and length of stay was numerically higher although not statistically significant in the therapeutically anticoagulated group. Although a lack of blinding may introduce some bias into these findings, there may not be a benefit in anticoagulating all allograft thrombi, and prospective studies are needed to delineate which need higher doses of heparin.

While there is a lot of unknown in this area, allograft thrombosis and rebleeding due to excess anticoagulation negatively affect a patient’s postoperative course. Prospective studies on careful titration of anticoagulation will hopefully reduce these problems and play an important role in an ERAS pathway for pancreas transplantation.

## 6. Immunosuppression

Immunosuppression post-transplantation can be split into two categories: induction therapy and maintenance therapy. Induction therapy is started intra-operatively or shortly afterwards with the main aim of reducing acute rejection episodes. Maintenance therapy consists of long-term oral medications for reducing the risk of rejection, often a combination of tacrolimus, mycophenolate mofetil and steroids. This review will focus on induction immunosuppression as this is given during the initial hospital stay, and optimising induction therapy will help in the immediate postoperative course. Gruessner et al. found that 90% of patients received some form of induction immunosuppression between 2010–2014, even though multiple RCTs have failed to demonstrate a specific benefit of induction immunosuppression [143,144]. These older RCTs excluded high-risk categories (Donation after cardiac death and panel reactive antibodies > 20%) and used IL-2 receptor antagonists (daclizumab or basiliximab), potentially explaining the lack of benefit. Similar to anticoagulation, many centres will have their own specific immunosuppression protocols based on local variables and preferences. This review will aim to identify themes that should be incorporated into an ERAS protocol, but the specifics must be verified by prospective randomised studies.

The main choice of induction therapy currently is anti-thymocyte globulin (ATG), alemtuzumab and basiliximab. Some B-cell depleting therapies have been reported but used very infrequently (<0.6%), and were not be the focus of this review. All three induction therapies have acceptable rates of rejection, with multiple observational and prospective studies finding very little difference between the three [143,144,145,146,147,148,149,150,151,152,153,154,155,156].

In a retrospective study, Reddy et al. found similar rates of acute rejection (12% vs. 15%), CMV infection (18% vs. 10%) and one-year pancreas allograft survival (94% vs. 88%) in groups treated with ATG and alemtuzumab respectively [149]. In a prospective single centre RCT comparing single-dose intra-operative alemtuzumab induction therapy with a multi-dose ATG regimen in SPK patients, Stratta et al. found similar five-year patient survival (82% vs. 89%), five-year pancreas allograft survival (64% vs. 55%) and similar five-year rejection rates (21% vs. 44%, *p* = 0.12) in alemtuzumab vs. ATG respectively. They found slightly higher rates of CMV infection in the ATG group (16% vs. 0%, ATG vs. alemtuzumab, *p* = 0.054) [151]. In a nine year follow up of a prospective randomised trial, Bosmuller et al. found no difference in survival between ATG and alemtuzumab induction therapy [147]. A large retrospective study analysing the outcomes from 6860 SPK transplant recipients in the US in 2002–2009 found that compared to no induction, alemtuzumab, IL-2 receptor antibodies (IL-2RAb), and ATG did not affect patient mortality, pancreas graft survival or kidney graft survival [152]. Alemtuzumab had decreased length of initial hospital stay compared to the other three groups (median LOS 8 days vs. 10 days vs. nine days vs. 10 days, alemtuzumab vs. none vs. Il-2RAb vs. ATG, *p* < 0.001) although all induction modalities were associated with higher 6-month hospitalisation rates than no induction at all.

With no clear benefit of any induction therapy, we suggest that if induction therapy is to be used, it should be focused on other non-clinical factors. Alemtuzumab can be given subcutaneously and is known to have a similar effect to IV administration [157], is cheaper than ATG and requires one to two doses. ATG, on the other hand, requires central access with infusion over at least four hours and multiple doses [156]. Alemtuzumab is known to be effective at reducing the risk of rejection and is possibly associated with a shorter hospital LOS. Given the lack of evidence pointing to the superiority of other induction methods and the demonstrable benefits of alemtuzumab, centres should use this as part of their ERAS pathway.

Finally, it remains unanswered in the literature as to which patients benefit and require induction therapy given the prospective and retrospective studies failing to find a long-term benefit. There is a belief that there exists a population of patients that are at higher risk of rejection who would benefit from induction therapy. Immunosuppression is a major cause of morbidity post solid organ transplantation, and personalised immunosuppression may help reduce complications. A scoring system accounting for the risks of rejection due to donor factors, recipient factors and intra-operative factors would allow for quantification of this risk and tailored immunosuppression. One specific area of interest is HLA immunogenicity calculating the exact immunological risk based on donor and recipient phenotypes [158]. There are many methods of quantifying this, with studies showing a possible benefit in both kidney and pancreas transplantation [159,160,161,162,163]. Immunogenicity is just one factor that should be accounted for, and the development of a scoring system to decide on the need for induction immunosuppression should be implemented in an ERAS pathway to reduce any unnecessary risks from induction therapy.

## 7. Postoperative ERAS Components

### 7.1. Postoperative Nutrition

Early enteral nutrition (EEN) has been a mainstay of ERAS protocols in abdominal surgery with reduced length of stay, reduced postoperative complications and improved survival in gastrectomies, liver transplants and surgery involving bowel anastomoses [164,165,166,167,168,169]. Although there have not been RCTs evaluating EEN in pancreatic transplantation, it remains in the EPSEN (European Society for Clinical Nutrition and Metabolism) guidelines. The main concern with early enteral nutrition in abdominal surgery surrounds the anastomoses. In this regard, pancreaticoduodenectomies (PD) are comparable to pancreas transplantation as they both involve small bowel anastomoses, and some centres will use a Roux-en-Y reconstruction for the exocrine drainage [170]. Although some trials have demonstrated increased complications with EEN in PD patients [171,172], multiple meta-analyses have confirmed that early enteral nutrition is associated with reduced length of stay, improved nutritional status and no significant difference in postoperative complications [173,174,175]. A retrospective study on EEN in SPK patients confirmed that EEN patients received better nutritional support with no change in 30-day mortality, one-year graft loss or hospital length of stay [176]. Another retrospective study showed that prolonged postoperative hypoalbuminemia in SPK patients is associated with increased graft loss and decreased survival [177]; although albumin is not a highly specific marker for malnutrition, there is evidence that EEN can increase postoperative albumin levels compared to other regimens [178,179]. Although the benefit of EEN has not been established in pancreas transplant patients, the clear, demonstrable benefit in other similar abdominal surgeries and safety data in SPK patients suggests that EEN should be in a pancreas transplant ERAS protocol.

### 7.2. Post-Operative Nausea and Vomiting (PONV)

Post-operative nausea and vomiting (PONV) is known to be a particularly unpleasant experience for the surgical patient, often resulting in dehydration and delayed discharge [180]. While there is little research on the prevention of PONV in pancreatic transplantation, there is much work in the field of abdominal and pancreatic surgery. Initially, we suggest attempting to identify those at high risk of PONV using a simple scoring system such as the Apfel score [181]. This system looked at four risk factors: female gender, history of PONV, non-smoker and use of postoperative opioids. With the previously mentioned multi-modal analgesia techniques, we aim to reduce the use of opioids, reducing the risk of PONV. In those with high baseline risk, further interventions to reduce PONV include peri-operative dexmedetomidine and propofol total intravenous anaesthesia (TIVA) [182,183]. The use of dexmedetomidine has been proven in multiple fields of surgery in prospective studies, and we recommend its use in pancreas transplantation for its analgesic benefits and reduction in PONV [111,184,185].

Those with two or more risk factors should receive routine combination prophylaxis with at least two anti-emetics. There exists much literature on this topic, including a very detailed consensus statement on this issue [182]. There are multiple options for anti-emetic agents, and we recommend a multimodal approach.

### 7.3. Gastroparesis

Gastroparesis, also known as delayed gastric emptying, is a common complication seen in diabetic patients, with some studies putting its prevalence at 40% in type 1 diabetics in tertiary care centres [186]. It is due to neuronal damage from chronic hyperglycaemia resulting in poor gastric contractions; this can lead to early satiety, nausea and vomiting [187]. Cerise et al. found symptoms of gastroparesis in 18% of pancreas transplant patients and that it was associated with a mildly longer LOS [186]. The range of interventions targeting gastroparesis can be split into preventative measures and treatment. Opioid use in those with gastroparesis is known to be associated with increased mortality and LOS [188]. We have discussed many ideas for reducing opioid use in the analgesia section, and these complications in gastroparesis highlight another benefit with reduced opioid use. Avoidance of NG tube placement is also known to be beneficial has been discussed in its own section above. For the treatment of gastroparesis, prokinetic agents including dopamine antagonists (metoclopramide and domperidone) and motilin agonists (erythromycin) may play a role. Multiple RCTs and a meta-analysis have shown their benefit in gastroparesis and critically ill patient intolerant of feeding [189,190,191,192]. Erythromycin is not the optimal choice for pancreas transplant recipients as it is known to be a CYP450 inhibitor that can increase tacrolimus levels [189]. Given the possible effect of discontinuing erythromycin on tacrolimus and the tachyphylaxis associated with this drug, we do not routinely recommend this in our ERAS pathway. Although metoclopramide has not been directly studied in pancreas transplantation, we believe its strong evidence for treatment of gastroparesis means that it should be used postoperatively in patients with gastroparesis, with some centres already using it routinely in all post-operative patients [186].

### 7.4. Postoperative Mobilisation

Significant emphasis is placed on early mobilisation and physiotherapy involvement post-surgery, and rightfully so. It has been well established that early ambulation, especially in high risks patient reduces postoperative complications and LOS [193,194,195,196]. A 23 h protocol for laparoscopic nephrectomy was developed in the early 2000s involving ambulation on the day of surgery itself; it was safe and effective [197,198,199]. In cardiac surgery, mobilisation has shown to increase lung function, reduce atelectasis and reduce pleural effusions [200]. Pulmonary complications are common post major surgery, and early mobilisation following abdominal surgery may provide a respiratory benefit. Multiple RCTs have shown that early mobilisation is feasible, safe and effective in critically ill patient and patients in the surgical intensive care unit, reducing their ICU stay, increasing their muscle strength, and increasing their readiness for discharge [201,202,203,204,205]. Finally, early drain removal may aid with mobilisation by reducing the number of tubes connected to the patient while appearing to have no clear impact on outcomes after pancreatic surgery [206,207,208,209,210]. A Cochrane review of prophylactic abdominal drainage in pancreatic surgery found no difference in 30-day mortality, LOS or post-operative complications [206]. They suggest that drain use might be associated with slightly reduced 90-day mortality. They suggest that active suction drainage systems might reduce LOS, but a more recent retrospective study of 3430 patients found an increased incidence of SSIs [211]. In pancreas transplantation, there is a single retrospective study showing the drain use is associated with reduced re-operation rates [211]. Due to the lack of research on prophylactic drainage in the pancreas transplant patient, we can neither recommend for or against their use nor a specific timeline for removal. We suggest that drains are removed as soon as feasible to allow for increased mobilisation. We also recognise the need for more work on the benefits of drainage and the timing of drain removal.

Similarly, early catheter removal may also be beneficial to pancreas transplant recipients. They are routinely inserted in renal transplant to allow for close monitoring of urine production, but multiple studies have shown that patients can safely tolerate catheter removal within 48 h of surgery [212,213,214,215]. Meta-analyses within other surgical specialities have demonstrated decreased incidence of urinary tract infections and retention with early removal [216,217]. While no studies have been carried in pancreas transplant patients, we think it is reasonable to recommend early removal of the urinary catheter in stable patients, given the studies in related surgical fields. Early mobilisation must be included in any ERAS pathway for pancreatic transplantation.

## 8. Future Directions

### Machine Perfusion of Grafts

Machine perfusion (MP) of organs is emerging as a new strategy to resuscitate and monitor graft function before implantation. There is evidence to suggest perfusion strategies offer benefits over static cold storage (SCS) in other organ transplants, but there is still further work to be done in the field of pancreas transplantation. In kidney transplantation, machine perfusion has been shown to decrease delayed graft function and increase creatinine clearance at one month, especially in extended criteria kidney grafts [218,219,220,221]. Two of these RCTs [220,221] found no difference in the LOS of a patient undergoing HMP vs. SCS, but in a Brazilian centre receiving kidneys with an average cold ischemic time of more than 22 h, they found a hybrid machine perfusion strategy reduces LOS (13 days vs. 18 days, *p* < 0.011, hybrid machine perfusion vs. SCS, respectively) [218]. A large multicentre RCT comparing normothermic machine perfusion (NMP) with SCS found a reduced discard rate in the NMP group but no difference in length of hospital stay or length of time in ICU [222].

Multiple in vitro studies have looked at optimising perfusion strategies for pancreata which, in contrast to kidneys, require a low flow low-pressure environment [223,224,225,226,227,228]. We are yet to determine the optimal timings, perfusion solutions or biomarkers for assessing graft viability. The most promising research is looking into delivering therapeutic medications to the graft while on machine perfusion; in vitro experiments have shown that antithrombotic agents during HMP reduce microvascular thrombotic sequelae in kidneys [229]. This could be particularly beneficial in pancreas transplantation as some studies report up to an 18% graft thrombosis rate [230]. At the moment, there is not enough evidence to call for the use of machine perfusion in an ERAS pathway for pancreas transplantation, but future prospective studies should investigate the impact of machine perfusion on graft function and patient outcomes. The use of therapeutic interventions may help prevent the feared complication of allograft thrombosis and has the potential to change the postoperative course in these patients.

## 9. A Proposed ERAS Pathway

We have summarised our list of proposals to consider for any program starting an ERAS pathway for pancreatic transplantation in Table 1. This list is not comprehensive but contains many of the unique aspects of this surgery that should be included in future pathways.

## 10. Conclusions

ERAS protocols are becoming widely adopted in centres across the world and are an evidence-based approach to improving the care of the surgical patient. Although each individual measure may only have a small effect, their combined effect is known to improve patient LOS, readmission rates, cost of admission and overall patient satisfaction. The unplanned and complex nature of pancreas transplantation make developing ERAS protocols challenging, but we as a field must address these challenges for the care of our patients. Further prospective studies are needed to confirm each element of the pathway, but we are confident that with this work, a multidisciplinary team effort will improve patient outcomes in pancreatic surgery.

## Figures and Tables

**Table 1 jcm-10-01418-t001:** Proposed enhanced recovery after surgery (ERAS) component for pancreatic transplantation.

ERAS Component	Description	Strength of Recommendation	Level of Evidence
Informed consent	Vital to any surgery to keep the patient educated and invested in the process	Strong	NA
Prehabilitation program	Physical exercise program while on the waiting list to improve cardiovascular reserve	Strong	1b
Weight loss advice	Aim to reduce postoperative complications due to obesity	Strong	3b
Cardiovascular assessment	Screening with electrocardiogram (ECG), transthoracic echocardiography (TTE), and functional and/or non-invasive imaging with the possibility of invasive coronary angiogram	Medium	3b
Iliac vessel assessment	Preoperative CT without contrast of iliac vessels to allow for operative planning	Strong	3b
Hypercoagulability screening	Preoperative screening to guide perioperative and postoperative anticoagulation	Strong	3b
Smoking cessation counselling	An effective program aimed to produce smoking cessation at least four weeks preoperatively	Strong	1a
Antibiotic prophylaxis	First generation cephalosporin prophylaxis with consideration of antifungal discontinued within 96 h	Strong	1b
Dexmedetomidine	Pre-induction single IV dose of dexmedetomidine for improved postoperative analgesia	Medium	1b
Graft position	Ipsilateral placement of kidney and pancreas graft in SPK transplantation if other recipient factors allow	Weak	3b
Operative instruments	Standardised procedure with the increased use of automated equipment to minimize operative time	Medium	5
Consideration of gastroduodenal artery (GDA) reconstruction	Reduces bleeding risk and graft perfusion	Weak	4
Machine perfusion	Possible future inclusion to rescue and monitor marginal grafts	Weak	5
Enteric drainage	Reduced postoperative complications	Strong	3b
Need for induction therapy	Risk assessment algorithm to decide if the patient is high risk and would benefit from induction therapy	Strong	3b
Alemtuzumab induction therapy	Effective therapy with reduced length of stay (LOS) not requiring central access	Medium	1b
Platelet function analysis	Preoperative analysis with Col-Epi and Col-ADP assays for calculation of arterial thrombosis risk	Weak	3b
Thromboelastography (TEG) analysis	Rapid analysis of coagulation allowing titration of postoperative anticoagulation	Medium	3b
Standardised grading of graft thrombosis	Procedure for grading thrombosis on CT imaging allowing for standardized management	Medium	4
Analgesia: local block	Transversus abdominis plane (TAP)/quadratus lumborum block inserted at the time of operation reducing opioid use and faster return of bowel function (ROBF)	Strong	1b
Paracetamol	Regular IV paracetamol in the immediate post-operative period as an adjunct to other analgesics	Strong	1b
Lidocaine patch	Consider topical 5% lidocaine patch to reduce pain around incision sites	Weak	1b
Fluid therapy	Avoid excessive fluid administration with more work needed before goal-directed fluid therapy (GDFT) can be recommended	Medium	3b
Antiemetics	Calculation of post-operative nausea and vomiting (PONV) risk and regular use of multimodal anti-emetics therapy based on risk	Strong	1a
Postoperative nutrition	Early enteral nutrition	Strong	1a
Nasogastric (NG) intubation	Avoid routine use of prophylactic NG intubation for gastric decompression	Strong	1a
Metoclopramide	Regular metoclopramide in patients with delayed gastric emptying	Strong	1b
Early removal of drains	Removal of prophylactic drains as soon as possible to improve mobilisation	Weak	5
Early removal of urinary catheter	Aim to remove urinary catheter within 48 h of surgery if patient stable	Medium	3b
Postoperative mobilization	Early on the day, mobilisation in intensive care settings	Strong	1a

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
