# Peer review of "Working towards an ERAS Protocol for Pancreatic Transplantation: A Narrative Review"

_jcm, 2021, doi:10.3390/jcm10071418_

Round 1
Reviewer 1 Report
This manuscript is a comprehensive review of Enhanced recovery after surgery (ERAS) protocols for pancreas transplantation.
Patients who are eligible for pancreas transplantation have a long history of diabetes and often with hemodialysis treatment. As the authors mentioned, performing pancreas transplantation for these patients as an emergency, and immunosuppressive therapy following invasive surgery, induced a high incidence of complications.
Therefore, the ERAS protocols described in this manuscript are essential for pancreas transplantation. This manuscript, which is well organized and educationally, covers preoperative evaluation and education for patients, surgery, perioperative management including immunosuppressant, and even postoperative management.
Reviewer 2 Report
Congratulations to the authors for an excellent narrative review on the development of a pancreatic transplantation ERAS pathway. The review comes to fill a significant gap in the current ERAS literature.
Main Comments:
Workup:
- BMI: is there a recommended BMI cutoff for pancreas transplantability, particularly for T2DM? This might be beyond the scope of this study
Peri-transplant:
Mirroring the structure of non-transplant pancreatic surgery ERAS protocols, the authors should perhaps elaborate further:
- Incision: it might be helpful to have a short discussion over the relevance of incision choice.
- Graft positioning: The authors explained their reasoning in support of the enteric drainage technique; is orientation ( "tail up" vs "down") and laterality ( right vs. left side) of any relevance?
- Post-op nausea: as in pancreatectomy ERAS protocols, delayed gastric emptying and post-op nausea is a common issue on labile diabetics; comments on this may be helpful ( ? pre-op dexamethasone, post-op metoclopramide, erythromycin?);
- NGT: yes/ no; early vs. late removal
- Post-op ileus: besides pain management, are there any more evidence-based recommendations on this front ( e.g. early discontinuation of IVF to limit bowel edema as in the colorectal and most other ERAS protocols) ?
- Po analgesia: any recommendations over non-opiate oral pain meds scheme ( e.g regular paracetamol, gabapentin, lidocaine patch etc.)?
- Foley: timing of removal
- Drains: as a pancreatic/ transplant surgeon-reader, I would be keen to hear more about the eternal questions: yes/ no, number, positioning and duration of keeping the drains, and criteria of having them removed
- I would recommend to provide with the level of evidence (I-V) for all recommendations listed on the table.
Once again- this has been an important work on a subject that has been missing from the international ERAS literature, prepared at a leading pancreatic transplant institution.
Reviewer 3 Report
The authors present a review of application of ERAS protocol to pancreas transplantation. This is a good well-rounded review of the different factors impacting perioperative pancreas transplant outcomes. ERAS protocols are being designed and adapted by various surgical disciplines. Pancreas transplant is a complex procedure with significant perioperative complications. Therefore, this is an important review of pre-/intra-/ postoperative factors impacting pancreas transplant perioperative outcomes and how to potentially improve them.
I do have a few questions about their recommendations:
- For smoking cessation pre-transplant, the evidence authors present is for smoking cessation for > 4 weeks, but their recommendation is for cessation >6 weeks. Is that an arbitrary value they chose or is there evidence to support that?
- Weight loss/ prehabilitation: Do the authors suggest a particular weight loss goal/ BMI target? As for prehabilitation, while it is ideal to get patients in a better shape before transplant, since some of pancreas transplant candidates have h/o amputations/ vision loss as a result of their diabetes, it may be a difficult target.
- Cardiac assessment: Based on the evidence provided, do the authors suggest getting CT coronary angiography for all pancreas transplant patients because of high incidence of coronary disease in this population and as the authors reported, 10% post-test probability of significant coronary disease despite normal DSE?
- GDA reconstruction: I don't think most pancreas transplant surgeons reconstruct GDAs. I know there is some data on this but I don't think it is robust and this should not be part of ERAS protocol.
- Immunosuppression: As the authors report, there are differing data and opinions in the transplant community about induction and choice of induction agent for pancreas transplantation. The studies comparing alemtuzumab vs thymo induction show similar graft outcomes. Alemtuzumab is easier to administer compared to Thymo but I don't think there is enough evidence showing reduced LOS with that. Therefore, I would say that induction can be done with either Thymo/ Alemtuzumab.
- Anticoagulation: In my experience, platelet function monitoring is rarely done and is of uncertain value. The anticoagulation protocols widely vary between programs, with some programs just giving aspirin to these patients, while others start some kind of anticoagulation intra-/post-op with either heparin/lovenox/dextran. Do the authors suggest including some kind of anticoagulation post-operatively? If yes, what agent?/ when to start anticoagulation?/duration of anticoagulation?
Overall, I commend the authors to have done a thorough review of factors impacting perioperative pancreas transplant outcomes and incorporating them in the proposed ERAS protocol.
